# Nitrogen and Silicon Contribute to Wheat Defense’s to *Pyrenophora tritici-repentis*, but in an Independent Manner

**DOI:** 10.3390/plants13111426

**Published:** 2024-05-21

**Authors:** Andrea Elizabeth Román Ramos, Carlos Eduardo Aucique-Perez, Daniel Debona, Leandro José Dallagnol

**Affiliations:** 1Departamento de Fitossanidade, Faculdade de Agronomia Eliseu Maciel, Universidade Federal de Pelotas, Pelotas 96010-900, Rio Grande do Sul, Brazil; aroman@ueb.edu.ec; 2Laboratory of Phytopathology, Agricultural Sciences Natural Resources and the Environment Faculty, Bolivar State University, Guaranda EC020150, Ecuador; 3Czech Advanced Technology and Research Institute, Palacký University, Šlechtitelů 27, CZ-78371 Olomouc, Czech Republic; carloseduardo.auciqueperez@upol.cz; 4Agronomy Department, Universidade Tecnológica Federal do Paraná—Campus Santa Helena, Santa Helena 85892-000, Paraná, Brazil; debona.daniel@gmail.com

**Keywords:** callose, hydrogen peroxide, necrotrophic pathogen, plant defense, tan spot

## Abstract

Nitrogen (N) and silicon (Si) are mineral elements that have shown a reduction in the damage caused by tan spot (*Pyrenophora tritici-repentis* (*Ptr*)) in wheat. However, the effects of these elements were studied separately, and the N and Si interaction effect on wheat resistance to tan spot remains elusive. Histocytological and biochemical defense responses against *Ptr* in wheat leaves treated with Si (+Si) at low (LN) and high N (HN) inputs were investigated. Soil amendment with Si reduced the tan spot severity in 18% due to the increase in the leaf Si concentration (around 30%), but it was affected by the N level used. The superoxide dismutase (SOD) activity was higher in +Si plants and inoculated with *Ptr*, leading to early and higher H_2_O_2_ and callose accumulation in wheat leaf. Interestedly, phenylalanine ammonia-lyase (PAL) activity was induced by the Si supplying, being negatively affected by the HN rate. Meanwhile, catalase (CAT), and peroxidase (POX) activities showed differential response patterns according to the Si and N rates used. Tan spot severity was reduced by both elements, but their interaction does not evidence synergic effects in this disease’s control. Wheat plants from −Si and HN and +Si and LN treatments recorded lower tan spot severity.

## 1. Introduction

Tan spot, caused by the necrotrophic fungus *Pyrenophora tritici-repentis* (Died.) Drechs (formerly known as *Drechslera tritici-repentis* (Died.) Shoem.) (*Ptr*), is a significant wheat disease worldwide [1]. In the main wheat-producing region of southern Brazil, tan spot stands out as the predominant leaf disease [2]. Recent studies of the *Ptr*-wheat interaction showed that the activation of defense responses differs with the level of susceptibility of wheat cultivars, and the biochemical and histocytological defense responses are more robust in genotypes with moderate resistance [3,4]. In moderately resistant cultivars, oxidative burst (H_2_O production in the first hour after inoculation) constitutes an efficient strategy to active plant defense mechanisms, resulting in the fast accumulation of hydrogen peroxide (H_2_O_2_) in the epidermal cells [5]. Overall, oxidative burst can induce higher activity of the antioxidative system which is governed by enzymatic activity, such as superoxide dismutase (SOD), catalase (CAT), and peroxidase (POX), as well as defense-related enzymes chitinase (CHI) and phenylalanine ammonia-lyase (PAL), this one leading to an early (<18 h after infection (hai)) and more intense fluorescence of epidermal cells due to the accumulation of phenylpropanoid derivatives at the infection site [3,4]. However, in the susceptible cultivar, although some of the same defense response was observed, they occurred later, such the accumulation of H_2_O_2_ in the epidermal and mesophyll cells as after 24 hai and the fluorescence in epidermal cells after 72 hai [3,4]. In the susceptible genotype, the accumulation of H_2_O_2_ during *Ptr* attack is probably the result of toxin action, which is known to be a pathogen-triggered response, inducing cell death in the host [1].

Nitrogen (N) and silicon (Si) have been mineral elements utilized successfully for disease control in crops [6,7]. In wheat plants, besides the use of moderately resistant cultivars, crop rotation, and fungicide-spraying, several studies have reported that N and Si fertilization are considered complementary alternatives for the management of Blast, Fusarium head blight, Powdery mildew, Spot blotch, Stripe rust, and tan spot [8,9,10].

Nitrogen is an essential element for plant growth and plays a key role in health and yield [11,12]. Several studies have shown contrasting effects of N fertilization on disease development. In some, N increases disease severity and reduces the incubation period, due to delay in lignin deposition, and reduces the thickness of the secondary cell walls, resulting in a higher severity level [6]. Nevertheless, other studies have shown decreased severity because an increase in N is probably associated with a healthy area duration (HAD) [13,14]. Regarding biochemical plant defense mechanisms, higher N led to lower activity of PAL and reduced concentration of C-based secondary metabolites such as some phytoalexins. On the other hand, there was a positive effect on pathogenesis related (PR) proteins, such as CHI and β-1,3-glucanase, along with increases in amino acid metabolism, which acts as a precursor to many defense-related phytoalexins, and increases in the activity of antioxidant enzymes [6]. Furthermore, N also influences other compounds involved in the defense against pathogens, such as the production of nitric oxide, polyamines, hormonal signaling molecules, apoplastic sugars and amino acids, but the effect is variable with the N source: nitrate or ammonium [6,15].

In turn, Si is considered a non-essential plant nutrient, although it plays a beneficial role in promoting plant growth and increasing yield by alleviating biotic and abiotic stresses in many plant species, especially plants belonging to Poaceae family [7,9,16]. As is well known, Si reduces disease intensity through physical and biochemical mechanisms [17]. The physical mechanism, restricting or delaying the penetration and spread of fungal hyphae in host tissues, occurs due to the formation of the Si-cuticle double layer, cell wall strengthening, and the formation of papillae [17,18,19]. Furthermore, plants supplied with Si showed priming of biochemical defenses through increased activity of defense enzymes, biosynthesis and accumulation of antimicrobial compounds (phenylpropanoids, flavonoids, phytoalexins, and PR proteins), and the accumulation of H_2_O_2_ at infection sites when infected by fungi, mainly [3,4,7,20,21,22]. Additionally, Si can influence the effect or movement of effectors, which, in turn, confers stronger plant resistance by keeping effectors from reaching the targets or the plant signals from being recognized by the pathogen [23].

Nitrogen and Si nutrition has shown significant increments in wheat diseases’ control [3,5,6,10,12,13,15,24]; however, few studies have hypothesized synergistic effects associated with N and Si supplementation in combination to reduce disease severity levels. Experiments carried out in wheat fields demonstrated that the presence of N and Si did not result in any significant effect on wheat production, Si absorption, or leaf rust disease development [25,26]. For this reason, we hypothesize that Si supply in combination with a higher N fertilization rate in wheat plants may develop a synergistic effect, benefiting a higher effective control of tan spot. Given the essential role of N in wheat production, understanding the intricate interactions between N and Si is crucial for unraveling their combined effects on plant disease. In this study, our aim was to explore and understand the role of Si and its interplay with nitrogen in eliciting histocytological and biochemical defense responses during *Ptr* infection in wheat.

## 2. Results

All variables and parameters analyzed from each experiment when submitted to Hartley’s test (F_max_ test) showed homogeneous variances, allowing for their combination.

### 2.1. N and Si Concentrations in Wheat Leaves Inoculated with Ptr

The Si leaf concentration and N leaf concentration showed a differential response according to the treatment applied. In this study, significant differences (*p* < 0.05) for Si input and N rate, as well as their interaction, Si × N, rates, were observed (Table 1).

The Si × N rates interaction showed that under +Si and HN, a reduction up to 21% in leaf Si concentration was observed, indicating that an increased N rate reduced the concentration of Si in leaf tissues (Table 2). On the other hand, similar values of N concentration occurred in wheat leaves regardless of Si input (Table 2).

### 2.2. Disease Intensity

The factor associated with the Si × N interaction was significant (*p* < 0.05) for FLS, NLs and SEV (Table 1). Plants treated with +Si and LN showed reductions of 35, 70 and 21% for FLS, NLs, and SEV, respectively, when compared to plants treated with −Si and LN (Table 2). Curiously, in plants treated with +Si, there was no difference for FLS regardless of N rate. Meanwhile, in treatments that receive −Si and HN, lower NLs and SEV values were observed than in plants +Si and HN (Table 2).

### 2.3. Effect of N Rates and Si on Callose Accumulation and on Hydrogen Peroxide Accumulation

The Si and N rates were significant (*p* < 0.05) for callose accumulation (Table 1). The greatest accumulation of callose was observed at 24 hai and 32 hai, after which it declined (Table 3; Appendix A).

Leaves from +Si plants showed higher accumulation of callose by up to 90% compared to −Si plants, especially with HN rate (Table 3). Regarding N rate, in leaves from plants with LN, there was 50% increase in the accumulation of callose compared with HN plants (Table 3). Furthermore, regardless of the N or Si input, the peak of callose deposition occurred at 32 hai (Figure 1A).

The DAB reaction, used for detection of H_2_O_2_, revealed that the intensity of the H_2_O_2_ accumulation varied over time, and the interaction of N and Si treatments was significant (*p* < 0.05) (Table 1; Appendix A). The intensity of H_2_O_2_ accumulation varied among the Si × N interaction (Table 1). The Si × N interaction showed that leaves from +Si and LN had a higher intense DAB reaction at HI, MI, and LI compared with −Si and LN plants (Table 4). With +Si and HN input and +Si and LN plants, there was a higher percentage of MI and HI reactions compared to −Si and LN plants. For −Si plants at any N rate, no significant differences occurred in the percentage of DAB reaction classification. However, +Si and HN decreased the percentages of DAB reactions classified as LI and MI, and increased those classified as HI (Table 4). LI and MI were increased at 48 hai in the treatment of +Si and LN rate (Figure 1B,C). The peak of H_2_O_2_ accumulation intensity (HI) was observed 48 and 60 hai in treatments with Si and high N rate (Figure 1D). Interestedly, in the treatment of +Si and LN rate, the peak of H_2_O_2_ accumulation intensity was observed at 16 and 24 hai, indicating rapid reaction in this treatment compared with that with +Si and HN (Figure 1D).

### 2.4. Effect of N and Si on Enzyme Activity

The Si × N interaction was significant (*p* < 0.05) for POX, SOD, and PAL activities, except for CAT (Table 1). The +Si and LN plants had higher activity of POX, SOD, and PAL compared to the −Si and LN plants (Table 5).

In this study, a higher activity up to 37, 2 and 9% of POX, SOD, and PAL, respectively, was observed in treatments with +Si and LN compared with treatments with −Si and LN (Table 5). Differently, for CAT activity, an alteration in the activity for each element was observed individually, being significant only for the N rate (*p* < 0.05) (Table 1). For example, HN-supplied plants showed a 60% increased CAT activity compared with plants without LN (Table 6). In the case of Si, no significant difference was observed among the treatments applied (Table 1).

In general, the SOD, CAT, POX, and PAL activities increased over time after plant inoculation, reaching the highest values at 48 hai, then remaining high until 72 hai for CAT and POX, with a slight decrease in SOD and PAL activities after 60 hai (Figure 2).

SOD activity in *Ptr*-inoculated plants was higher in +Si and LN input, from 8 to 48 hai, and from 16 to 32 hai under +Si and HN input, compared to −Si and HN, with the highest value being reached at 32 hai. In −Si and LN plants, the SOD activity also increased over time, although more slowly than in +Si and LN plants, reaching the highest activity at 72 hai. The SOD activity in −Si and LN plants reached a value similar to +Si and LN at 60 hai, and the highest activity occurred at 72 hai, while under −Si and HN input, the value was similar to that of +Si/HN at 48 hai (Figure 2A).

For the CAT activity in *Ptr*-inoculated plants, higher activity occurred with LN input at 24 and 48 hai in +Si and LN plants compared to −Si/LN ones, but in −Si and LN plants, higher CAT activity was recorded at 60 and 72 hai (Figure 2B). Under −Si/HN input, CAT activity increased over time, peaking from 24 to 48 hai, compared to +Si and HN or +Si and LN. In the case of POX, the activity started to increase eight hours earlier in −Si plants than in +Si ones under both N rates (Figure 2C). However, in +Si and LN, the POX activity reached the highest value at 60 hai and remained high up to 72 hai, and achieved the peak of activity at 48 hai, 24 h earlier than the peak observed for the −Si and HN input (Figure 2C).

The PAL activity increased over 60 to 72 hai under +Si and LN and up to 48 hai with +Si and HN input, being significantly higher compared to −Si and LN ones until 60 hai (Figure 2D). Under −Si/HN, the PAL activity was significantly greater from 8 to 32 hai, compared to +Si and HN or +Si and LN (Figure 2D).

### 2.5. Principal Component Analysis (PCA)

The principal component analysis (PCA) showed that for treatments of Si and N rates, the two first PCs explained 83.8% (PC 1 = 57.7%; PC 2 = 26.1%) of the total variation (Figure 3A).

Biochemical variables and leaf Si concentration correlated negatively with N concentration (PC 1), while disease variables (FLS, NLs, and SEV) correlated positively and highly associated with PC 2. FLS, NLs, and SEV showed high association with −Si and LN plants, while +Si and LN plants had a high correlation with variables such as leaf Si concentration, PAL activity, LI and CAL mainly (Figure 3A). On the other hand, −Si and HN and +Si and HN plants showed a negative association with biochemical variables such as SOD, CAL, CAT, and MI (Figure 3A). The leaf’s N concentration had a negative correlation with PAL, CAT, and POX activities, as well as LI and MI. Meanwhile, PAL activity, leaf Si concentration, MI, HI, and LI were highly correlated. POX and CAT activities showed high correlation with MI (Figure 3B).

## 3. Discussion

In this study, wheat plants supplied with Si and N in combination displayed a differential response in resistance components to *Ptr*, affecting plants’ histocytological and biochemical defense responses. A higher Si concentration in wheat plants stimulated earlier and higher H_2_O_2_ accumulation, callose deposition, and increased the enzymatic activity of metabolic pathways related to the antioxidative and defense systems, which together caused a reduction in tan spot severity. In addition, the high N concentration interfered with the Si-beneficial action on the plant defense metabolism.

The metabolic benefits of Si supply in plants with low N input were better evidenced compared to higher N input, resulting in earlier and stronger activation of defense responses, indicating a beneficial role of Si with low availability of N in potentiating the plant defenses against *Ptr*. A higher SOD activity indicated that +Si plants respond rapidly to pathogen infection, which is associated with increases in the H_2_O_2_ concentration, resulting in a reduction in NLs per cm². This observation was also reported by Dorneles et al. [3], who found that earlier accumulation of H_2_O_2_ in wheat epidermal cells was associated with the plants’ defense mechanism in Si-supplied plants and in cultivars with moderate resistance, as opposed to the later accumulation of H_2_O_2_, and mainly in the parenchymal cells, in susceptible cultivars. Likewise, the increase in the activity of POX and PAL observed in +Si plants under low N input indicated that the defense mechanism involving the phenylpropanoid metabolism was activated, such as lignin biosynthesis [27], which may contribute to reducing the lesion size and the final disease severity. PAL is a key enzyme of the phenylpropanoid pathway, which leads to increased synthesis of resistance-related phenols [28]. Previous studies also reported that phenylpropanoid derivatives and lignin accumulation have important roles in wheat defense against *Ptr* [3,4]. Interestingly, in plants under low N input, Si nutrition stimulated SOD, POX and PAL activities, starting them earlier and with greater activity, while under high N input, although SOD activation was similar to that of low N input, the changes in the activity of POX and PAL occurred a little later or at a lower intensity. However, there was a reduction in CAT activity, indicating different strategies under high and low N input in the potentiation of defenses by Si leading to an increase in the H_2_O_2_ and phenylpropanoid derivatives.

Regarding PAL activity, studies have reported that in plants receiving low N input, its activity increases, but only slightly, while at high N input, the activity decreases [29]. The results of our study partially corroborate that finding; although low and high N inputs increased PAL activity in response to *Ptr*, under high N, the change in the enzyme activity was slightly delayed. Despite this difference in PAL activity at high N input, there was less *Ptr* damage as indicated by the lower tan spot severity in this study and as was previously reported by Castro et al. [30] and Fleitas et al. [24]. In this sense, at high N, although rapid accumulation of H_2_O_2_ was induced, PAL activity was not the first response to counteract the infection, although it contributed to delayed colonization, inducing resistance to *Ptr*.

Nevertheless, observations in the enzymatic defense system, H_2_O_2_ accumulation, and callose deposition may be associated with the effect of +Si. Callose deposition is promoted by indole glucosinolates (IGSs) and reactive oxygen species (ROS) [31]. Furthermore, the activation of abscisic acid (ABA) signaling induces or primes callose deposition and is influenced by carbohydrate metabolism [31,32]. In our study, callose deposition began at 24 hai, reaching the highest peak at 32 hai in plants supplied with +Si/LN input. Indeed, highest callose and H_2_O_2_ accumulation were observed in +Si leaves infected by *Ptr* regardless of N input. For callose formation, a glucose molecule is required [33], resulting in higher carbon sources that probably come from a sucrose metabolism. In this context, sucrose metabolism plays a pivotal role in the cellular function and biosynthesis of starch, cellulose, callose, and proteins through hexoses and their derivatives produced by the metabolic pathway. Thus, sucrose metabolism stands as a cornerstone process, sustaining the energetic and biosynthetic demands of cellular life [34]. This observation agrees with the fact that Si impact the source–sink relationship and stimulates amino acid remobilization observed in rice [35]. For instance, the wheat–*Pyricularia oryzae* interaction mediated by Si nutrition improved the source–sink relationship of infected leaves due to an alteration in the activities of the enzymes’ acid invertase and sucrose phosphate synthase in leaves and spikes of wheat challenged by *Pyricularia oryzae* [36], suggesting that in our case, Si incorporation, besides contributing to severity reduction, may perhaps involve maintaining the sucrose’s concentration despite the *Ptr* infection and without a direct influence of N concentration.

Associated with this observation, Si concentration in leaves decreased with a high N rate. At least two factors may have influenced this outcome: first, the greater increase in biomass with higher N, which led to the dilution of Si in the plant tissue; and second, changes in the soil pH caused by the higher N rate affected Si uptake. In this context, soil acidification under high N rates in this experiment may have affected Si absorption, which mainly occurs through active transport mechanisms inherent in the roots [37], resulting in lower Si content at high N. This observation agrees with Murozuka et al. [38], who reported that differences in leaf Si concentration among wheat genotypes were partially explained by differences in soil pH.

This is the first report of the effect of N × Si interaction in wheat challenged by *Ptr*. At low N, the Si concentration was higher in the plants, demonstrating the effect on *Ptr* due to enhancement in plant defenses. The findings of this study suggest that Si and low N input increase callose deposition in a compatible reaction (susceptible host). There were no synergistic effects between the two elements in resistance to tan spot. Despite an effective control of *Ptr* at high N, N fertilization should be used with caution when Si is incorporated in the soil during crop management. Nitrogen application and soil amended with Si are strategies that activate plant defense mechanisms independently because increasing the N rate affects Si concentration, altering the changes in the plants’ defense responses to *Ptr*. In conclusion, the Si and N input treatment altered the activities of SOD, CAT, POX, and PAL, improving the defense against *Ptr* infection. In addition, the greater accumulation of H_2_O_2_ and deposition of callose in leaves infected by *Ptr* suggests that the activation of these defenses was mainly associated with Si.

## 4. Materials and Methods

### 4.1. Plant Material and Growth

Wheat cultivar TBIO Tibagi (BIOTRIGO GENETICA^®^, Passo Fundo/RS—Brazil), susceptible to tan spot, was used in the experiment. Seeds were surface-sterilized in 10% (vol vol^−1^) NaOCl for 2 min and rinsed in sterile water for 3 min. Pre-germination was conducted in wet/dry cycles of 12 h, at 20 °C, in the dark for 2 days. Then, seven germinated seeds were transferred to biodegradable seed nursery bags (14 × 18 cm) (Huvai^®^, USA) containing 475 g of soil and 100 g of seed starter (Carolina Soil^®^, Santa Cruz do Sul/RS, Brazil), and kept in plastic trays (47 × 25 × 10 cm), ten in total. The plants were grown in a greenhouse with relative humidity of 80 ± 5% and temperature of 25 ± 2 °C.

### 4.2. Soil Characteristics, Silicon Amendments, and Nitrogen Treatment

The soil used in this experiment was collected in the experimental area of the Federal University of Pelotas, Capão do Leão, Rio Grande do Sul, Brazil. The collected soil had the following physicochemical characteristics: 658 g kg^−1^ sand; 241 g kg^−1^ clay; 101 g kg^−1^ silt; 1% organic matter; 1.5 g N kg^−1^; 6.64 g NO_3_ kg^−1^; 1.93 g NH_4_ kg^−1^; 11.9 kg P; 143 mg K dm^−3^; 2.7 cmolc Ca dm^−3^; 1.4 cmolc Mg dm^−3^; 0.9 mg Cu dm^−3^; 0.8 mg Zn dm^−3^; 12.3 mg Mn dm^−3^; pH 5.5; cation exchange capacity (CEC), 10.4 cmolc dm^−3^; H + Al, 0.3 cmolc dm^−3^; and base saturation, 48.1%.

The concentration of available Si (extracted with 0.01 M CaCl_2_) was 6.0 mg dm^−3^, determined according to the standard colorimetric analysis method [39].

The Si source was calcium silicate (Agrosilício^®^, Agronelli Insumos Agrícolas, Uberaba, Brazil), which is composed of 10.5% Si, 25.0% Ca and 6.0% Mg. Calcium silicate was mixed with the soil at an equivalent rate of 13.2 tons ha^−1^ to increase the soil’s pH to 6.0. To isolate the effect of Si, we standardized the amount of Ca and Mg supplied to the plants in the calcium silicate treatment. The extra-fine limestone (Dagoberto Barcelos, Caçapava do Sul, Brazil), composed of Ca (26.5%) and Mg (15%) magnesium, was added to the soil at an equivalent rate of 11 tons ha^−1^, being utilized for plant growth in the control treatment. Calcium carbonate (Synth, Diadema, Brazil) and magnesium carbonate (Synth, Diadema, Brazil) were used to adjust the final concentrations of Ca and Mg, respectively, between treatments. After complete soil homogenization of each treatment, water was added to reach 80% of field capacity, followed by 30 days of incubation in plastic bags, until use.

The nutritional requirements of wheat plants were supplied using an adjusted nutrient solution of phosphorus and potassium. The nutrient solution contained 12.97 g L^−1^ of KH_2_PO_4_ and 15.62 g L^−1^ of KCl. A volume of 63.5 mL of this nutrient solution described above was applied to each bag at the third and fourth weeks after transplanting. Also, a N requirement was applied as a nutrient solution adjusted to obtain the two doses under study, low N (LN): 70 kg ha^−1^, and high N (HN): 200 kg ha^−1^. The two N concentrations were prepared containing 0.147 g L^−1^ and 0.442 g L^−1^ for LN and HN, respectively, using granular urea (N = 45%). A volume of 40 mL of the N solution was applied per bag, according to the N rate planed, in the fifth to seventh weeks. Deionized water was utilized to prepare both nutrient solutions and plant irrigation.

### 4.3. Experimental Design and Treatments

Two experiments were carried out under a completely randomized experimental design in a factorial arrangement of 2 × 2, consisting of two Si treatments [not supplied (11.0 ton of limestone; −Si) and supplied (13.2 ton of calcium silicate; +Si)] and two N rates (LN:70 or HN:200 kg N ha^−1^), with twelve replications. In each experiment, the fourth and fifth leaves (from the top), at a phenological stage of 37 according to Zadoks Growth Scale, of each plant were marked and used for pathometrical, histocytological, and biochemical analyses. Each experimental unit corresponded to a plastic bag containing seven plants.

### 4.4. Inoculum Production and Inoculation Procedure

The *Pyrenophora tritici-repentis* race 1 (BRPtr8), provided by Universidade de Passo Fundo, was used in the experiments. The fungus, preserved as a PDA mycelial disk at −20 °C, was reactivated on PDA for one week, after which mycelial plugs with a diameter of 5 mm were cut from the margin of actively growing colonies and transferred to Petri dishes containing modified V8-agar medium [(3.0 g L^−1^ of calcium carbonate (Synth, Diadema, Brazil), 150 mL L^−1^ of tomato sauce (Fugini^®^, Monte Alto/SP, Brazil), 15 g L^−1^ of agar (Kasvi, Curitiba, Brazil)] and distilled water.

The inoculum production was performed according to the method described by Dorneles et al. [3]. Briefly, after fungal growth in the modified V8-agar medium over five days at 25 ± 1 °C in darkness, it was stressed by mycelium scraping. Then, the fungal colony was exposed for 24 h to light at 25 ± 1 °C to allow for the development of conidiophores, followed by a further 24 h of darkness at 15 ± 1 °C, necessary for conidia formation. After seven days, conidia were carefully removed from the Petri dishes with a soft bristle brush using water containing 1 drop of Tween 20. The conidial suspension was adjusted to obtain a concentration of 3 × 10^3^ conidia mL^−1^ as described by Dorneles et al. [3].

The fourth leaves of 50-day-old wheat plants with the flag leaf just visible (ZGS37) were used in the experiment. For this, leaves were picked and immediately placed in NaOCl at 10% (vol vol^−1^) for 2 min, followed by washing three times with sterilized water and kept in water for 3 min, and air drying for 5 min. The conidial suspension was applied with a hand-held sprayer (Tecblas, REF: 60 mL/Porto Alegre, Brazil) on the adaxial surface of the leaves. After inoculation, six leaves were transferred to a plastic Petri dish (150 × 15 mm) with water–agar medium. Mock-inoculated leaves (control) were sprayed with distilled water and exposed to the same conditions as *Ptr*-inoculated ones. The plastic Petri dishes with *Ptr*-inoculated or mock-inoculated leaves were incubated at 25 ± 1 °C, relative humidity of 80 ± 5%, and a photoperiod of 12 h during the experiment.

### 4.5. Disease Assessment

Disease assessments were performed at 96 h after inoculation (hai) by quantifying the number of lesions per cm^2^ (NLs), as well as determining the final lesion size (FLS) and disease severity (SEV). For NLs, four areas (1 cm^2^) of six leaves were used to quantify the NLs formed. The final lesion size (FLS) was quantified at four points of one cm^2^ of leaf by randomly measuring at least five lesions with a digital caliper (150 mm, model 02KC4PR7P1995, YuanSen^®^, Shenzhene, China). The SEV, defined as the percentage of total leaf area affected by the disease (necrotic and chlorotic tissue), was determined in six inoculated leaves per replicate. For this, digital leaf images were obtained at 600 dpi using a scanner (Epson/L395), and the SEV was determined using the digital image analysis software Quant^®^ (Universidade Federal de Viçosa, Viçosa/MG, Brazil).

### 4.6. Leaf Samples for Detection of Hydrogen Peroxide and Aniline Blue Staining

To determine if N rate and Si supply altered the amount and/or time of hydrogen peroxide accumulation and callose deposition in response to pathogen presence, *Ptr*-inoculated leaves were sampled at 8, 16, 24, 32, 48, 60, 72 and 96 hai. To verify that hydrogen peroxide or callose was produced in response to the presence of the pathogen, the analysis and image acquisition were only performed after verifying the presence of the pathogen (conidia and germ tube or appressoria) in the plant tissue. Additionally, to prove that the defense responses were only due to inoculation with the pathogen, leaf samples from mock-inoculated plants were sampled at the same time as *Ptr*-inoculated plants. However, due to the absence of pathogen structures on these plants, they were used only as a control to verify that the alteration in defense responses was not due to environmental factors or manipulation of the leaf tissue.

#### 4.6.1. Detection of Hydrogen Peroxide

Hydrogen peroxide in the plant tissue was detected according to the method described by Shi et al. [26]. Briefly, fragments of 1 cm^2^ were cut from the middle of each leaf, discarding the tip and base of the leaf, and immediately vacuum infiltrated in diaminobenzidine (DAB) (Sigma-Aldrich, Jurubatuba/SP, Brazil ) solution (wt vol^−1^) for 15 min (1 mg mL^−1^, pH 3.8, adjusted using 0.1 N HCl), which was freshly made in 10 mM phosphate buffer (pH 5). Leaf samples were incubated in the solution at room temperature and exposed to light (fluorescent tube, Osram 40 W) for 8 h. Finally, the leaf samples were discolored in boiling 70% ethanol. After this period, the leaf samples were kept in 5 mL of 70% ethanol in Falcon tubes for further documentation.

##### Microscopic Analysis, DAB Image Acquisition, and Quantification of Affected Epidermal Cells

For microscopic analysis, 16 leaf samples of 1 cm^2^ per treatment were examined at each sampling time. Leaf samples were placed adaxial side up on slides with one drop of 20% glycerol and covered by a coverslip, followed by observation under a CX41 biological microscope (Olympus^®^, Shinjuku-ku/Tokyo, Japan) at 200× magnification. To verify that hydrogen peroxide was produced in response to the presence of the pathogen, image acquisition was only performed after verifying the presence of the pathogen (conidia and germ tube or appressoria). The number of epidermal cells showing brown staining (DAB polymerized), indicating the presence of H_2_O_2_, was counted at each infection site. For each sampling time, at least 20 appressorial sites were randomly observed and photographed for further analyses. Leaf image acquisition was performed using a digital camera with 3.1 MP (USB 2.0 Color CMOS Digital Eyepiece Microscope Camera, AmScope, Irvine/CA, USA) linked to the PC via a USB interface. AmScope 4.11 software was used as an interface for DAB image acquisition. All photographs were acquired with 2038 × 1536 resolution and saved as JPEG files.

After this, a set of reference images with a visible H_2_O_2_ reaction at the infection site caused by *Ptr* were selected to determine the intensity profile (value between 0 and 255 RGB). For this, the ‘magic wand’ and histogram tools of the Adobe Photoshop CS6^®^ 13.0 v64 software were used to manually select visible H_2_O_2_ reactions and determine the intensity scale (IS) based on a previous categorical color intensity scale according to Dorneles et al. [3]. In this way, the IS was established considering three color intensities: low intensity or slightly brownish wall of the epidermal cells [(LI); range between 111 and 131 RGB)]; medium intensity or wall of epidermal cells encircled by a dark brownish color [(MI); range between 91 and 110 RGB)]; and high intensity or dark brown entire epidemic cells [(HI); values ≤ 90 RGB)]. Based on the IS from the training images, 12 images for each treatment were evaluated. The histogram analysis tool of ImageJ/Fiji 1.46 was used, which allows for the attainment of an intensity profile for RGB images. For each image, the number of pixels was quantified according to the IS. At the end of these analyses, the IS pixels of the image evaluated were expressed as a percentage of H_2_O_2_ reaction based on the ratio between the total number of pixels at each IS and the total number of pixels into which each image was decomposed.

#### 4.6.2. Aniline Blue Staining

To detect callose, plant tissue was stained with aniline blue (Sigma-Aldrich, Jurubatuba/SP, Brazil) as described by Schenk and Schikora [27], with some modifications. Briefly, leaf samples (1 cm^2^) were discolored in 1:3 [acetic acid/ethanol 95% (vol vol^−1^)]. The saturated destaining solution was replaced every 12 h, if necessary, until all tissues were transparent. Then, leaf samples were washed in 0.07 mM phosphate buffer (pH = 9) for 30 min. Next, the destained leaf samples were incubated in the dark for at least 8 h in a staining solution containing 0.001 mg mL^−1^ aniline blue fluorochrome (wt vol^−1^) (Sigma-Aldrich, Jurubatuba/SP, Brazil) in 0.07 mM phosphate buffer (pH = 9), and kept in Falcon tubes. After incubation, the leaf samples were washed again in 0.07 mM phosphate buffer (pH = 9), placed in 70% ethanol for fixation, and finally embedded in 50% glycerol (vol vol^−1^) in the dark for conservation. After this procedure, leaf segments at each sampling time were placed and fixed with the adaxial side up on slides containing drops of Entellan^TM^ new rapid mounting medium (Sigma-Aldrich, Jurubatuba/SP, Brazil) with a drop of 50% glycerol (vol vol^−1^), and maintained in the dark for further microscopic analysis.

##### Microscopic Analysis, Image Acquisition, and Callose Quantification

For microscopic analysis, 16 leaf samples of 1 cm^2^ per treatment at each sampling time were visualized for callose deposition through epifluorescence microscopy (Eclipse Ts2-FL, Nikon, Melville/NY, USA) using a DAPI filter (wavelength of 385 nm and maximum emission of 420 nm). The Capture 2.2 software was used for documentation. The images were acquired at 1300 × 1030 resolution with adjustment of brightness at 300 ms exposure time, and saved as JPEG files. As for hydrogen peroxide, to ensure that the callose was produced in response to the presence of the pathogen, image acquisition was only performed after verifying the presence of the pathogen at the infection site.

Once again, a set of reference photographs was used to determine the intensity profile (value between 0 and 255 RGB). For this purpose, the ‘magic wand’ and histogram tools of Adobe Photoshop CS6^®^ were used to manually select callose intensity, which ranged between 139 and 176 RGB. Based on the intensity range identified from the training images, 12 images for each treatment were evaluated using the histogram analysis tool ImageJ/Fiji 1.46, which quantifies the number of pixels and intensity profile at the range of each RGB image. The callose deposition area was recorded as the corresponding pixels of range intensity (139 and 176 RGB) of the total number of pixels into which each image was decomposed and expressed as a percentage of callose based on the ratio between the number of callose pixels and the total number of image pixels.

### 4.7. Enzyme Activity

Inoculated leaves (*Ptr*-inoculated and mock-inoculated) were collected at 8, 16, 24, 32, 48, 60, 72 and 96 hai, flash-frozen using liquid nitrogen, and subsequently preserved at −80 °C until further examination. For each treatment and each sampling time, four samples were collected. Each leaf sample (300 mg) was ground with a mortar and pestle and the resulting fine powder analyzed for enzyme activity. The crude extract used for enzyme activity determination was obtained according to the method described by Dallagnol et al. [40].

The activities of catalase, peroxidase and superoxide dismutase were determined according to the methods described by Dallagnol et al. [40]. The activity of catalase (CAT, EC 1.11.1.6) was determined by quantifying the degradation of hydrogen peroxide (H_2_O_2_) (Merck, Pinheiros/SP, Brazil) and expressed as micromoles of H_2_O_2_ degraded min^−1^ mg^−1^ of protein. Peroxidase activity (POX, EC 1.11.1.7) was determined based on the colorimetric quantification of pyrogallol (Sigma-Aldrich, São Paulo, Brazil) oxidation, and expressed as moles of purpurogallin produced min^−1^ mg^−1^ of protein using an extinction coefficient of 2.47 mM cm^−1^. The superoxide dismutase (SOD, EC 1.15.1.1) activity was estimated based on the colorimetric quantification of the photoreduction of nitroblue tetrazolium (NBT) (Sigma-Aldrich, Jurubatuba/SP, Brazil). The specific activity of SOD was expressed in units of SOD mg^−1^ of protein, considering that one unit of SOD was the amount required to inhibit the photoreduction of NBT by 50%. Phenylalanine ammonia-lyase (PAL, EC 4.3.1.5) activity was determined according to the method described by Dorneles et al. [4] by colorimetric quantification of transcinnamic acid formed from phenylalanine (Sigma-Aldrich Jurubatuba/SP, Brazil) and expressed as µmol of transcinnamic acid produced per min^−1^ mg^−1^ of protein.

For each enzyme activity, four separate extractions of samples from each treatment were performed. Each extraction was read in a spectrophotometer (model UV-UM51, Bel Engineering srl^®^, Monza/MB, Italy) in the corresponding wavelength [CAT: 240 nm; POX: 420 nm; SOD: 560 mn; PAL: 290nm]. The soluble protein concentrations of the extracts were measured by the standard Bradford method, using bovine serum albumin as the standard protein wavelength [595 nm].

### 4.8. Analyses of N Leaf Concentration

Leaf samples (four) were collected from the just visible flag leaves (ZGS37) for each treatment, dried at 60 °C for 72 h, and ground using an R-TE-350 mill. A subsample of 1.0 g was used for determination of leaf concentration (g kg^−1^) according to the standard micro-Kjeldahl method.

### 4.9. Analyses of Si Leaf Concentration

Four leaves were collected at the same stage mentioned above for the leaf Si concentration (g kg^−1^) determination for each treatment. The samples were rinsed with deionized water, dried for 72 h at 65 °C, and ground to pass through a 40-mesh screen using a mill. The foliar Si concentration was determined by standard colorimetric analysis from 0.1 g of alkali-digested tissue [39].

### 4.10. Data Analyses

The data of callose, hydrogen peroxide and enzymatic activity were used to calculate the area under the curve (*AUC*) according to the equation of Shaner and Finney (1977).
$$AUC=\underset{i=1}{\sum }n\left(\frac{\left[Yi+n+Yi\right]}{2}\right)\left[Xi+1-Xi\right]$$
where *Yi* = values (per replication) in each observation; *Xi* = is the time (days) of each observation; and *n* = total number of observations. The Shapiro–Wilk test was applied to ascertain the normality of the data and the homogeneity of variances was checked with Bartlett’s test. Data obtained from two experiments were analyzed using Hartley’s test (*F*_max_ test) to determine the homogeneity of variances between experiments and could be combined. The data were analyzed by applying parametric ANOVA, and the means were compared by *F* and Tukey’s test (*p* ≤ 0.05). Principal components analysis (PCA) was used to determine the relationship among the variables and parameters. Statistical analyses were conducted with R version 4.0.4. through the packages “dplyr”, “FactoMiner”, “factoextra”, “ggplot2”, “ggcorrplot”, “ggpubr”, and “tidyr” [41].

## Figures and Tables

**Figure 1 plants-13-01426-f001:**
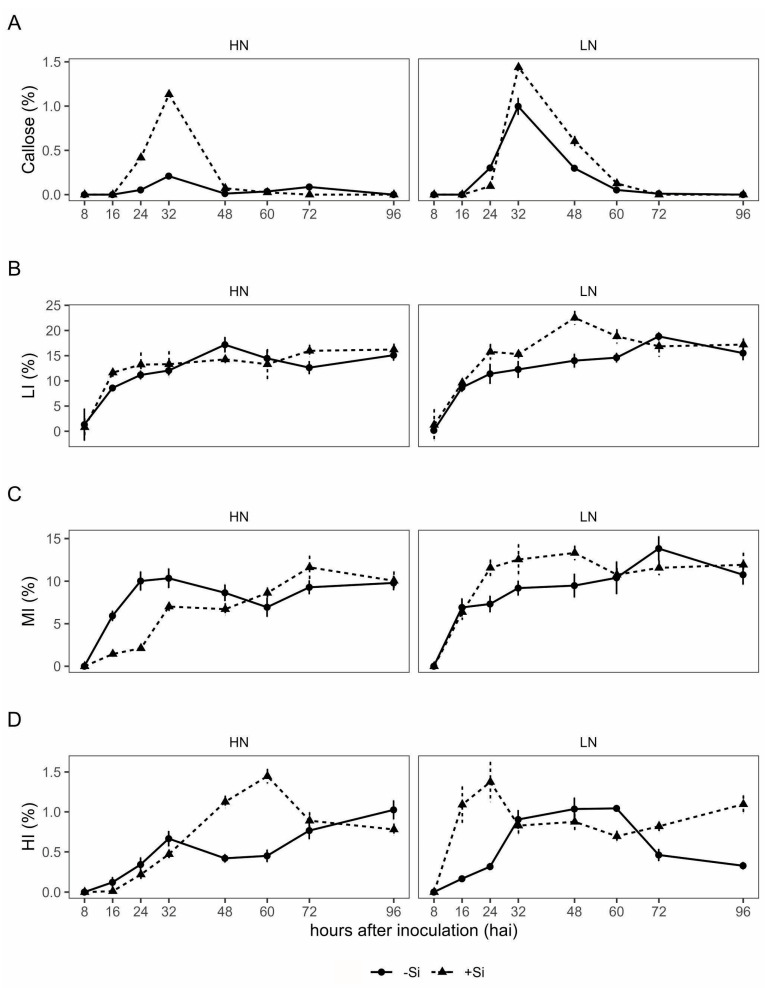
Callose deposition (**A**) and DAB reaction classified as low intensity (LI) (**B**), medium intensity (MI) (**C**) and high intensity (HI) (**D**) at hours after inoculation (hai) in the leaves of wheat plants grown in soil containing limestone (equivalent to 11.0 tons ha^−1^; −Si) or calcium silicate (equivalent to 13.2 tons ha^−1^; +Si) and low nitrogen (70 kg ha^−1^; LN) and high nitrogen (200 kg ha^−1^; HN) rates and inoculated with *Pyrenophora tritici-repentis*. Bars represent the standard deviation; *n* = 24.

**Figure 2 plants-13-01426-f002:**
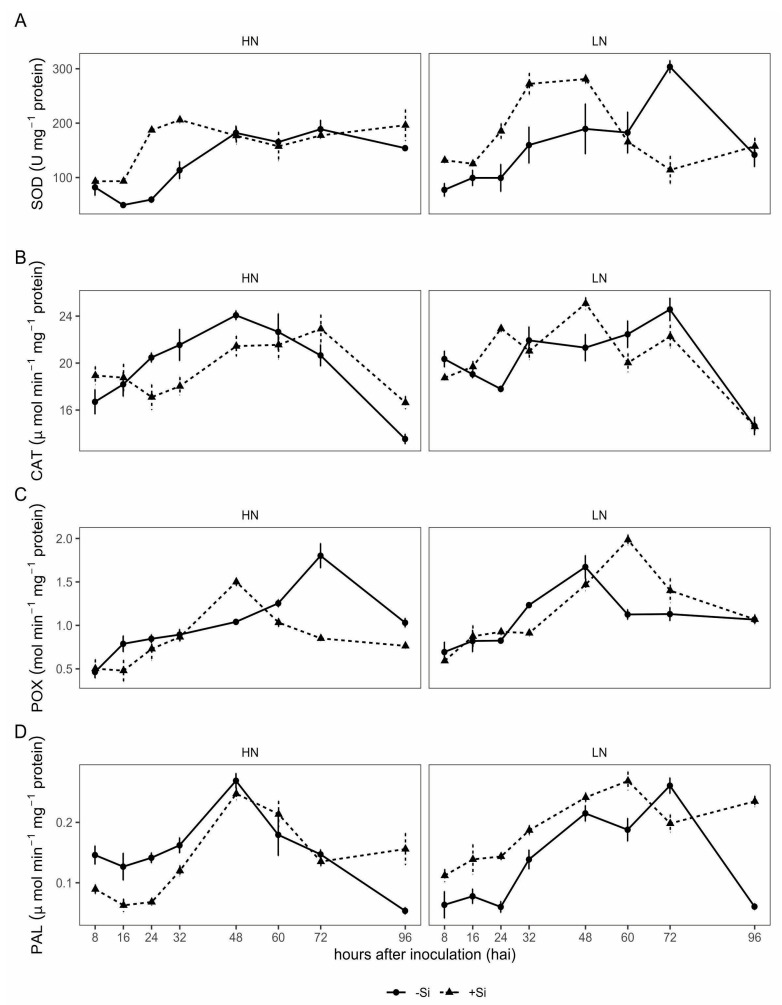
Activity of superoxide dismutase (SOD) (**A**), catalases (CAT) (**B**), peroxidases (POX) (**C**), and phenylalanine ammonia-lyase (PAL) (**D**) at hours after inoculation (hai) in the leaves of wheat plants grown in soil limestone (equivalent to 11.0 tons ha^−1^; −Si) or calcium silicate (equivalent to 13.2 tons ha^−1^; +Si) and low nitrogen (70 kg ha^−1^; LN) and high nitrogen (200 kg ha^−1^; HN) rates and inoculated with *Pyrenophora tritici-repentis*. Bars represent the standard deviation; *n* = 8.

**Figure 3 plants-13-01426-f003:**
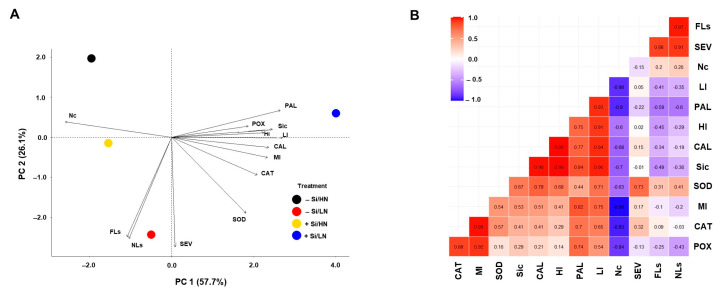
Biplot of loading values in the principal components analysis (PCA). Silicon (Si) × nitrogen rate (NR) (**A**) and correlation analyses (**B**) for variables and parameters from final lesion size (FLS), number of lesions (NLs) per cm^2^, severity (SEV) of tan spot, DAB reaction classified as low intensity (LI), medium intensity (MI), and high intensity (HI), and callose deposition in the leaves of wheat plants grown in soil containing limestone (equivalent to 11.0 tons ha^−1^; −Si) or calcium silicate (equivalent to 13.2 tons ha^−1^; +Si) and low nitrogen (70 kg ha^−1^; LN) and high nitrogen (200 kg ha^−1^; HN) rates and inoculated with *Pyrenophora tritici-repentis*.

**Table 1 plants-13-01426-t001:** Analysis of variance of the effects of silicon (Si) and nitrogen rates (N), and their interaction for Si concentration in leaf tissue (Leaf Si), N concentration in leaf tissue (Leaf N), disease severity (SEV), number of lesions per cm^2^ (NLs), final lesion size (FLS), area under low intensity of visible H_2_O_2_ reactions curve (LI_AUC_), area under medium intensity of visible H_2_O_2_ reactions curve (MI_AUC_), area under high intensity of visible H_2_O_2_ reactions curve (HI_AUC_), area under callose concentration curve (CAL_AUC_), area under catalase activity curve (CAT_AUC_), area under phenylalanine ammonia-lyase activity curve (PAL_AUC_), area under superoxide dismutase activity curve (SOD_AUC_), and area under peroxidase activity curve (POX_AUC_).

Variable/Parameter	*p* Value		
	Si	N	Si × N
Leaf Si	**<0.001**	**<0.001**	**<0.001**
Leaf N	**<0.001**	**<0.001**	**<0.001**
SEV	0.58	**<0.001**	**<0.001**
NLs	0.168	0.227	**<0.001**
FLS	**0.013**	0.131	**<0.001**
LI_AUC_	**<0.001**	**<0.001**	**<0.001**
MI_AUC_	0.950	**<0.001**	**<0.001**
HI_AUC_	**<0.001**	**<0.001**	**<0.001**
CAL_AUC_	**<0.001**	**<0.001**	0.062
CAT_AUC_	0.569	**<0.001**	0.207
PAL_AUC_	**<0.001**	**<0.001**	**<0.001**
SOD_AUC_	**<0.001**	**<0.001**	**<0.001**
POX_AUC_	**<0.001**	**<0.001**	**<0.001**

Bold values are significant at 5%.

**Table 2 plants-13-01426-t002:** Mean values of Si concentration in leaf tissue (leaf Si), N concentration in leaf tissue (leaf N), disease severity (SEV), number of lesions per cm^2^ (NLs) and final lesion size (FLS) in the leaves of wheat plants grown in soil containing limestone (equivalent to 11.0 tons ha^−1^; −Si) or calcium silicate (equivalent to 13.2 tons ha^−1^; +Si) and low nitrogen (70 kg ha^−1^; LN) and high nitrogen (200 kg ha^−1^; HN) rates and inoculated with *Pyrenophora tritici-repentis*.

Variable	Treatment				
	−Si		+Si		*n*
	LN ^z^	HN ^z^	LN ^z^	HN ^z^	
Leaf Si	18.16 c	16.9 d	25.2 a	20.6 b	8
Leaf N	29.4 b	31.9 a	32.9 a	26.0 c	8
SEV	16.2 a	10.9 c	12.8 b	13.8 b	24
NLs	13.9 a	7.5 b	7.3 b	11.4 a	24
FLS	0.98 a	0.74 b	0.70 b	0.82 b	24

^z^ Means within a row followed by the same lowercase letter are not significantly different (*p* ≤ 0.05) as determined by analysis of variance and Tukey’s test. *n* = repetitions.

**Table 3 plants-13-01426-t003:** Mean values of the area under callose concentration curve (CAL_AUC_) in the leaves of wheat plants grown in soil containing limestone (equivalent to 11.0 tons ha^−1^; −Si) or calcium silicate (equivalent to 13.2 tons ha^−1^; +Si) and low nitrogen (N) (70 kg ha^−1^; LN) and high nitrogen (200 kg ha^−1^; HN) rates and inoculated with *Pyrenophora tritici-repentis*.

		Si	
Parameter	N ^z^	−Si ^z^	+Si ^z^
CAL_AUC_	LN	24.4 Ba	60.6 Aa
	HN	8.6 Bb	37.5 Ab

^z^ Means within each column followed by the same lowercase letter or means within a row followed by the same uppercase letter are not significantly different (*p* ≤ 0.05) as determined by analysis of variance and Tukey’s test; *n* = 24.

**Table 4 plants-13-01426-t004:** Mean values of the area under low intensity of visible H_2_O_2_ reactions curve (LI_AUC_), area under medium intensity of visible H_2_O_2_ reactions curve (MI_AUC_), and area under high intensity of visible H_2_O_2_ reactions curve (HI_AUC_) in the leaves of wheat plants grown in soil containing limestone (equivalent to 11.0 tons ha^−1^; −Si) or calcium silicate (equivalent to 13.2 tons ha^−1^; +Si), and low nitrogen (70 kg ha^−1^; LN) and high nitrogen (200 kg ha^−1^; HN) rates and inoculated with *Pyrenophora tritici-repentis*.

Variable	Treatment				
	−Si		+Si		*n*
	LN ^z^	HN ^z^	LN ^z^	HN ^z^	
LI_AUC (×1000)_	1.2 b	1.1 c	1.4 a	1.2 b	24
MI_AUC (×100)_	8.6 b	7.4 c	9.6 a	6.4 d	24
HI_AUC_	54.0 c	49.0 d	78.1 a	66.2 b	24

^z^ Means within a row followed by the same lowercase letter are not significantly different (*p* ≤ 0.05) as determined by analysis of variance and Tukey’s test. *n* = repetitions.

**Table 5 plants-13-01426-t005:** Mean values of the area under phenylalanine ammonia-lyase activity curve (PAL_AUC_), area under superoxide dismutase activity curve (SOD_AUC_), and area under peroxidase activity curve (POX_AUC_) in the leaves of wheat plants grown in soil containing limestone (equivalent to 11.0 tons ha^−1^; −Si) or calcium silicate (equivalent to 13.2 tons ha^−1^; +Si), and low nitrogen (70 kg ha^−1^; LN) and high nitrogen (200 kg ha^−1^; HN) rates and inoculated with *Pyrenophora tritici-repentis*.

Variable	Treatment				
	−Si		+Si		*n*
	LN ^z^	HN ^z^	LN ^z^	HN ^z^	
PAL_AUC_	13.7 b	13.9 b	18.0 a	13.2 b	8
SOD_AUC (×1000)_	15.8 ab	12.3 c	16.1 a	15.0 b	8
POX_AUC_	100.8 b	100.1 b	110.1 a	79.9 a	8

^z^ Means within a row followed by the same lowercase letter are not significantly different (*p* ≤ 0.05) as determined by analysis of variance and Tukey’s test. *n* = repetitions.

**Table 6 plants-13-01426-t006:** Mean values of the area under catalase activity curve (CAT_AUC_), in the leaves of wheat plants grown in soil containing limestone (equivalent to 11.0 tons ha^−1^; −Si) or calcium silicate (equivalent to 13.2 tons ha^−1^; +Si) and low nitrogen (70 kg ha^−1^; LN) and high nitrogen (200 kg ha^−1^; HN) levels inoculated with *Pyrenophora tritici-repentis*.

		Si	
Parameter	N ^z^	−Si ^z^	+Si ^z^
CAT_AUC (×1000)_	LN	18.3 Aa	18.4 Aa
	HN	17.8 Ab	17.5 Ab

^z^ Means within each column followed by the same lowercase letter or means within a row followed by the same uppercase letter are not significantly different (*p* ≤ 0.05) as determined by analysis of variance and Tukey’s test; *n* = 8.

## Data Availability

The data that support the findings of this study are openly available as a compendium at https://osf.io/jpbk5/, (accessed on 1 February 2024), registered with the DOI 10.17605/OSF.IO/JPBK5.

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
