# Peer review of "Nitrogen and Silicon Contribute to Wheat Defense’s to Pyrenophora tritici-repentis, but in an Independent Manner"

_plants, 2024, doi:10.3390/plants13111426_

Round 1
Reviewer 1 Report
Comments and Suggestions for Authors
Manuscript entitled “Nitrogen and silicon contribute to wheat defense’s to Pyrenophora tritici-repentis, but in an independently manner” is of significant importance in context to the role of nitrogen and Si fertilization in plant growth and disease resistance. Overall MS is well prepared, however it needs to be revised to improve the quality.
- Check for similarity and reduce it throughout MS.
- Title: in an independently manner OR in an independent manner is better?
- Abstract needs to be completely revised. Reduce similarity, expand it overall and add numerical results in concluding lines.
- Line 53: Introduction; avoid starting sentences with abbreviations …..
- Lines 69-81. Beneficial role of silicon along with N fertilization and its impact on plant N uptake and finally plant growth can be expanded with suitable references on wheat, cereals and other crops ….. following reference may also be useful for some information https://doi.org/10.3390/agronomy12020413
- Lines 323-329: why only one susceptible cultivar was used in this study and not any resistant cultivar was assessed along with it?
- Lines 368-393: Correct and replace all degree symbols in temperature units, check throughout MS.
- How many samples were analyzed for each treatment for enzyme activities?
- Leaf N concentration: only one subsample per treatment was analyzed for leaf N concentration?
- Line 534: Cite the R package used for PCA and correlation plot
Comments on the Quality of English LanguageModerate editing of English language required
Author Response
Dear Reviewer, thank you for the comments. They improved the manuscript. We answer point-by-point below.
Reviewer 1
Manuscript entitled “Nitrogen and silicon contribute to wheat defense’s to Pyrenophora tritici-repentis, but in an independently manner” is of significant importance in context to the role of nitrogen and Si fertilization in plant growth and disease resistance. Overall MS is well prepared, however it needs to be revised to improve the quality.
- Check for similarity and reduce it throughout MS.
Authors: Checked
- Title: in an independently manner OR in an independent manner is better?
Authors: adjusted
- Abstract needs to be completely revised. Reduce similarity, expand it overall and add numerical results in concluding lines.
Authors: adjusted
- Line 53: Introduction; avoid starting sentences with abbreviations …..
Authors: adjusted
- Lines 69-81. Beneficial role of silicon along with N fertilization and its impact on plant N uptake and finally plant growth can be expanded with suitable references on wheat, cereals and other crops ….. following reference may also be useful for some information https://doi.org/10.3390/agronomy12020413
Authors: Added.
- Lines 323-329: why only one susceptible cultivar was used in this study and not any resistant cultivar was assessed along with it?
Authors: Resistant cultivars are not available to tan spot, only partial resistance. Previous studies demonstrated that Si can improve resistance to tan spot in both susceptible and moderate resistant cultivars. Then, we decided to use only a susceptible cultivar to reduce the size of the experiments, a situation that allows a better accuracy in the evaluations.
- Lines 368-393: Correct and replace all degree symbols in temperature units, check throughout MS.
Authors: adjusted
- How many samples were analyzed for each treatment for enzyme activities?
Authors: four samples per treatment in each sampling time. Information added in the manuscript.
- Leaf N concentration: only one subsample per treatment was analyzed for leaf N concentration?
Authors: four samples. Information added in the manuscript.
- Line 534: Cite the R package used for PCA and correlation plot
Authors: information added
Comments on the Quality of English Language: Moderate editing of English language required
Authors: adjusted

Reviewer 2 Report
Comments and Suggestions for Authors
Pyrenophora tritici-repentis is a necrotrophic fungus that causes important wheat disease worldwide, known as tan spot. Research on ways to reduce the incidence of the disease without the use of fungicides is of great importance. Research on many diseases caused by numerous pathogens shows that one of such possibilities is the use of silicon. On the other hand, there are conflicting reports about the effect of using nitrogen. The authors of the current work aimed to explain the role of silicon and understand the mechanisms of its action in increasing the resistance of wheat and to investigate the interaction with nitrogen by using its high and low doses. The authors conducted numerous experiments and obtained very valuable results for science and wheat disease management. The experimental results were well documented in tables and figures. However, there are places in the text that are unclear because either there are unfinished sentences or incorrect abbreviations are used. Examples are provided in Remarks. Remarks also provides examples of a few typographical errors. Introduction and Discussion are very interesting and comprehensively written. After carefully reading the manuscript and making corrections, the manuscript should be published in Plants.
Remarks
Line 20 – consider revising this sentence
Line 28 induced or caused?
Line 41 hai - the abbreviation should be explained when used for the first time in the text
Line 54-57 consider revising this long sentence
Line 82 incomplete sentence
Line 94 (Ptr) – this should be in line 29
Line 98 ‘on wheat leaves’ ?? or ‘in wheat leaves’ (see Table1 and Table 2)
Line 100-101 incomplete sentence
Line 110 -113 should have a distance under Table 1
Line 110 overcomplicated sentence
Line 111 Si, not ‘Sic’
Line 112-113 it is unclear what this means 'Also, in - Si/LN or -Si/LN'
Line 113 Language?
Line 132 Table 3 - there is no explanation of what does NL in Table 3 mean
Line 134 you write in the explanation to Table 3 'high nitrogen (200 kg ha-1; HN)' - where are the data for HN in Table 3?
Line 139 ‘especially with HN rate (Table 3).’ – where is the data for HN in Table 3?
Line 138-143 the entire paragraph is unclear - as a result of incorrect markings in Table 3?
Line 187 are inconsistencies in the explanations to Table 6 and the data in Table 6. Therefore, the text is unclear
Line 195 it should be (Fig. 2). This should also be corrected elsewhere in the text
Line 436 "at least 20 appressorial sites were randomly observed and photographed for further analyses" - unfortunately, the text does not contain the results of these observations, this requires explanation
Line 416, 426, 456… the section numbering is incomprehensible, if it is (Line 426) 4.6.1.1 then it should also be at least 4.6.1.2,
Line 566 “plant disease., American’ – it needs small correction
Line 589 tritici – it should be in italic
Line 610 Septoria tritici – it should be in italic
Comments on the Quality of English Languagesee Remarks
Author Response
Dear Reviewer, thank you for the comments. They improved the manuscript. We answer point-by-point below.
Reviewer 2
Pyrenophora tritici-repentis is a necrotrophic fungus that causes important wheat disease worldwide, known as tan spot. Research on ways to reduce the incidence of the disease without the use of fungicides is of great importance. Research on many diseases caused by numerous pathogens shows that one of such possibilities is the use of silicon. On the other hand, there are conflicting reports about the effect of using nitrogen. The authors of the current work aimed to explain the role of silicon and understand the mechanisms of its action in increasing the resistance of wheat and to investigate the interaction with nitrogen by using its high and low doses. The authors conducted numerous experiments and obtained very valuable results for science and wheat disease management. The experimental results were well documented in tables and figures. However, there are places in the text that are unclear because either there are unfinished sentences or incorrect abbreviations are used. Examples are provided in Remarks. Remarks also provides examples of a few typographical errors. Introduction and Discussion are very interesting and comprehensively written. After carefully reading the manuscript and making corrections, the manuscript should be published in Plants.
Remarks
Line 20 – consider revising this sentence
Authors: adjusted
Line 28 induced or caused?
Authors: adjusted (caused)
Line 41 hai - the abbreviation should be explained when used for the first time in the text
Authors: adjusted
Line 54-57 consider revising this long sentence
Authors: adjusted
Line 82 incomplete sentence
Authors: adjusted
Line 94 (Ptr) – this should be in line 29
Authors: adjusted
Line 98 ‘on wheat leaves’ ?? or ‘in wheat leaves’ (see Table1 and Table 2)
Authors: adjusted
Line 100-101 incomplete sentence
Authors: adjusted
Line 110 -113 should have a distance under Table 1
Authors: adjusted
Line 110 overcomplicated sentence
Authors: adjusted
Line 111 Si, not ‘Sic’
Authors: adjusted
Line 112-113 it is unclear what this means 'Also, in - Si/LN or -Si/LN'
Authors: adjusted
Line 113 Language?
Authors: adjusted
Line 132 Table 3 - there is no explanation of what does NL in Table 3 mean
Authors: adjusted
Line 134 you write in the explanation to Table 3 'high nitrogen (200 kg ha-1; HN)' - where are the data for HN in Table 3?
Authors: adjusted
Line 139 ‘especially with HN rate (Table 3).’ – where is the data for HN in Table 3?
Authors: adjusted
Line 138-143 the entire paragraph is unclear - as a result of incorrect markings in Table 3?
Authors: adjusted
Line 187 are inconsistencies in the explanations to Table 6 and the data in Table 6. Therefore, the text is unclear
Authors: adjusted
Line 195 it should be (Fig. 2). This should also be corrected elsewhere in the text
Authors: adjusted
Line 436 "at least 20 appressorial sites were randomly observed and photographed for further analyses" - unfortunately, the text does not contain the results of these observations, this requires explanation
Authors: These sites were evaluated for DAB and Callose quantification. These data are presented in tables 3 and 4, and figure 1. Supplementary material was added to illustrate these sites.
Line 416, 426, 456… the section numbering is incomprehensible, if it is (Line 426) 4.6.1.1 then it should also be at least 4.6.1.2,
Authors: adjusted
Line 566 “plant disease., American’ – it needs small correction
Authors: adjusted
Line 589 tritici – it should be in italic
Authors: adjusted
Line 610 Septoria tritici – it should be in italic
Authors: adjusted

Reviewer 3 Report
Comments and Suggestions for Authors
Dear Authors
The current manuscript entitled “Nitrogen and silicon contribute to wheat defense’s to Pyrenophora tritici-repentis, but in an independently manner” discuss
N-Si interaction effect on wheat resistance to tan spot and results suggested that +Si and high N input contributed to reduced tan spot severity, Si and N may act independently to activate plant defense responses. The research is well planned and presented nicely, although there are certain opportunities for further improvement. Please find my suggestions below.
Ø Please include the source of Wheat cultivar TBIO Tibagi (Biotrigo®)?
Ø The experiment was conducted in nursery bags in green house conditions? Please include details of number of bags, plants per bag and replicates per treatment.
Ø Line 340- “Calcium silicate was mixed with the soil at a rate of 13.2 tones ha-1 to increase the soil pH to 6.0.” please make it clear the application was done in field and then the soil was collected? Or soil was collected first and treated with Si? If so, how the soil was prepared? Field capacity? And sieved to remove the granules etc?
Ø Line 363- two Si treatments [not supplied (-Si) or supplied (+Si)] on disease assessment, please include the supplied Si amount.
Ø Line 381- The conidial suspension was adjusted to obtain a concentration of 3 × 103 conidia mL-1, is this concentration of conidia is enough for symptoms development? The amount seems less, please confirm.
Ø Line 399- “The SEV, defined as the percentage of total leaf area affected by the disease (necrotic and chlorotic tissue), was determined in six inoculated leaves.” The number of replicates are too less, especially in in-vitro experiments there must be more number of replicates.
Ø Line 395- The lesions were quantified after 96 hai? The 96 hr are usually the initiation of symptoms, which may develop completely after 1 week of inoculation. Is there any data after one week of inoculation?
Ø Detection of hydrogen peroxide in leaves- which leaves and age of the seedlings/plants?
Ø Please include the microscopic pictures of conidia for quantification and fungal growth plates.
Ø Line 426- include the microscopic images, may be as supplementary material.
Thank you
Regards
Author Response
Dear Reviewer, thank you for the comments. They improved the manuscript. We answer point-by-point below.
Reviewer 3
The current manuscript entitled “Nitrogen and silicon contribute to wheat defense’s to Pyrenophora tritici-repentis, but in an independently manner” discuss
N-Si interaction effect on wheat resistance to tan spot and results suggested that +Si and high N input contributed to reduced tan spot severity, Si and N may act independently to activate plant defense responses. The research is well planned and presented nicely, although there are certain opportunities for further improvement. Please find my suggestions below.
Ø Please include the source of Wheat cultivar TBIO Tibagi (Biotrigo®)?
Authors: included.
Ø The experiment was conducted in nursery bags in green house conditions? Please include details of number of bags, plants per bag and replicates per treatment.
Authors: information included.
Ø Line 340- “Calcium silicate was mixed with the soil at a rate of 13.2 tones ha-1 to increase the soil pH to 6.0.” please make it clear the application was done in field and then the soil was collected? Or soil was collected first and treated with Si? If so, how the soil was prepared? Field capacity? And sieved to remove the granules etc?
Authors: Soil were first collected, then chemical composition determinate, for later treatment application at equivalent rate. This information was added in the manuscript.
Ø Line 363- two Si treatments [not supplied (-Si) or supplied (+Si)] on disease assessment, please include the supplied Si amount.
Authors: information included.
Ø Line 381- The conidial suspension was adjusted to obtain a concentration of 3 × 103 conidia mL-1, is this concentration of conidia is enough for symptoms development? The amount seems less, please confirm.
Authors: Yes, the concentration is low, but enough to cause disease under experimental conditions.Drecheslera tritici-repentis did not produce much conidia under controlled environments, which make producing inoculum a difficult task.
Ø Line 399- “The SEV, defined as the percentage of total leaf area affected by the disease (necrotic and chlorotic tissue), was determined in six inoculated leaves.” The number of replicates are too less, especially in in-vitro experiments there must be more number of replicates.
Authors: We used six inoculated leaves per replicate. We made this information clear in the manuscript.
Ø Line 395- The lesions were quantified after 96 hai? The 96 hr are usually the initiation of symptoms, which may develop completely after 1 week of inoculation. Is there any data after one week of inoculation?
Authors: Tan Spot symptoms generally appear around 72 hours after inoculation, depending on the susceptibility of the host, but defense response, including H2O2 and callose deposition use to appear earlier than symptoms appearance. For this reason, we determine 96 hours in these experiments as the time to determine the disease severity.
Ø Detection of hydrogen peroxide in leaves- which leaves and age of the seedlings/plants?
Authors: information included.
Ø Please include the microscopic pictures of conidia for quantification and fungal growth plates.
Authors: picture included as supplementary.
Ø Line 426- include the microscopic images, may be as supplementary material.
Authors: picture included as supplementary.

Round 2
Reviewer 1 Report
Comments and Suggestions for Authors
Thank you for revising the Manuscript and providing the required information. Please reduce similarity index in the first two lines of Abstract, Headings 4.4. Inoculum production and inoculation procedure and Heading 4.7. Enzyme activity.
Comments on the Quality of English LanguageMinor editing of English language required
Author Response
Dear Reviewer,
Thank you for your consideration. Your recommendations were adjusted whenever possible to guarantee the full understanding of the text.
Reviewer 3 Report
Comments and Suggestions for Authors
Dear Authors
Thank you for providing the revised version and inclusion of all suggestions.
The manuscript has been improved significantly and i do not have any further query.
Regards
Author Response
We thank you for the time dedicated by the reviewer.